# AGENTIC 3D SCENE GENERATION WITH SPATIALLY CONTEXTUALIZED VLMS

*A dystopian set design reminiscent of Blade Runner 2049*

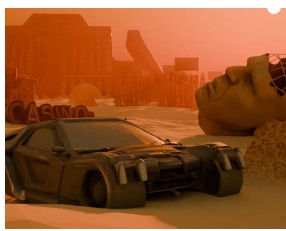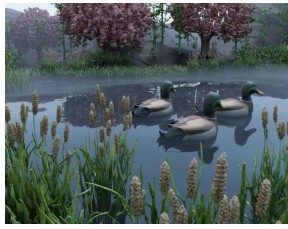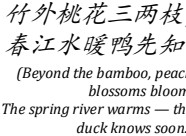

竹外桃花三两枝，
春江水暖鸭先知。

*(Beyond the bamboo, peach blossoms bloom,
The spring river warms — the duck knows soon.)*

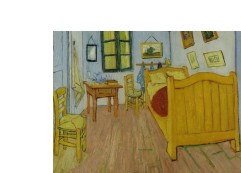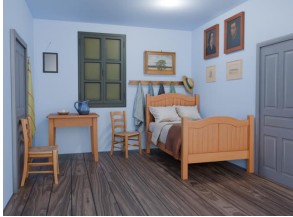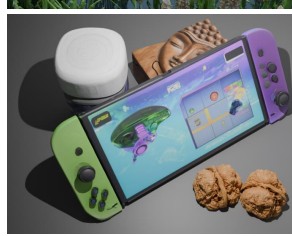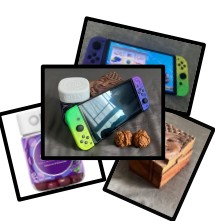

Figure 1: **Spatially contextualized VLMs.** We present an agentic 3D scene generation framework that augments VLMs with a structured spatial context. Our method supports diverse inputs—including text prompts, single images, and unstructured image collections—and produces coherent, editable, and semantically aligned 3D environments across varied styles and settings.

## ABSTRACT

Vision-language models (VLMs) have advanced multimodal generation, but extending them to structured 3D scene construction requires addressing three challenges: (1) integrating diverse inputs into a unified semantic–geometric representation, (2) capturing object–object and object–environment relations for layout reasoning, and (3) ensuring accurate and controllable 3D asset reconstruction. In response to these challenges, we introduce a framework for agentic 3D scene generation with **spatially contextualized VLMs**. The VLM first constructs a *spatial context* from multimodal user inputs, consisting of a scene portrait and a scene hypergraph. The scene portrait encodes semantic blueprints of the layout, objects, and environment, while the hypergraph captures unary, binary, and higher-order spatial relations. Injected with this structured context, the VLM performs *agentic 3D scene generation*, including asset synthesis, environment setup, layout planning, and ergonomic adjustment. Throughout generation, the spatial context is continuously read and updated whenever the VLM performs an operation, ensuring that the evolving scene remains coherent and semantically aligned. To further ensure coherent, editable, and semantically aligned 3D environments, we introduce an *auto-verification mechanism* that continuously monitors and corrects the scene during generation. This mechanism enforces fidelity to semantic constraints, geometric accuracy, and object–environment consistency. Experiments demonstrate strong generalization across diverse inputs and show that spatial context injection empowers VLMs with downstream capabilities such as interactive scene editing and path planning, advancing spatially intelligent systems in graphics and 3D vision.

## 1 INTRODUCTION

Recent advances in multimodal generation highlight the capabilities of large-scale vision-language models (VLMs) in interpreting and producing text, images, and video. Models like GPT-4o show strong performance in cross-modal reasoning, grounding, and language understanding. However, extending VLMs from 2D content to structured 3D scene construction poses unique challenges. Unlike images or videos, 3D scenes must maintain spatial consistency, ensure physical plausibility, and preserve semantic coherence. These demands require structured awareness of geometry and relations that current VLMs do not provide out-of-the-box.

3D scene construction requires solving three critical challenges: (i) integrating diverse multimodal inputs into a unified semantic representation that encodes the global layout, environmental setup, and object-level constraints, (ii) capturing object–object and object–environment interactions to guide spatial reasoning, and (iii) enabling controllable and accurate 3D asset generation with fine-grained placement and appearance.

In response to these challenges, we introduce a framework for *agentic 3D scene generation with spatially contextualized VLMs* (Figure 2). The VLM first constructs a **spatial context** from multimodal user inputs, consisting of two components:

- A *scene portrait*, which encodes semantic blueprints of the layout, objects, and environment. It integrates descriptive text and visual references into a high-level semantic representation of the scene.
- A *scene hypergraph*, which models unary, binary, and higher-order spatial relations, including ergonomic constraints, capturing both object–object and object–environment interactions.

Injected with this structured context, the VLM performs agentic 3D scene generation, including *asset synthesis*, *environment setup*, *layout planning*, and *ergonomic adjustment*. Throughout generation, the spatial context is continuously read and updated whenever the VLM performs an operation, ensuring that the evolving scene remains coherent and semantically aligned.

To further ensure fidelity, we introduce an **auto-verification mechanism** that continuously monitors and corrects the scene during generation. This auxiliary agent enforces semantic consistency, geometric plausibility, and object–environment coherence, providing iterative feedback to refine the generation process. Together, these modules allow the VLM to operate agentically: reading from, reasoning over, and updating the spatial context to produce coherent, editable, and semantically aligned 3D environments.

In our experiments, we demonstrate that this framework generalizes across diverse and challenging inputs, including natural language prompts, artistic references, photographs, and unstructured image collections. Compared with state-of-the-art approaches, our method produces semantically consistent 3D worlds that respect both object-level appearance and global layout coherence. Furthermore, injecting spatial context into VLMs empowers them to support a range of downstream tasks, such as interactive scene editing and path planning, advancing spatially intelligent systems in graphics and 3D vision. In summary, our key contributions are:

- We propose spatially contextualized VLMs that act as agents for structured 3D scene generation by constructing and maintaining a continually updatable spatial context.
- We design a **spatial context** composed of a *scene portrait* for multimodal integration and a *scene hypergraph* for relational reasoning, supporting layout planning and ergonomic adjustment.
- We introduce an **agentic scene generation** process that combines asset synthesis, environment setup, layout optimization, and ergonomic adjustment, supported by an **auto-verification mechanism**.
- We demonstrate that spatial context injection enables coherent, editable, and semantically aligned 3D environments across diverse inputs, while unlocking downstream capabilities such as interactive scene editing and path planning.

## 2 RELATED WORK

**3D Scene Generation.** Generating coherent 3D scenes with multiple objects is fundamentally more challenging than single-object synthesis, as it requires modeling both detailed geometry and

global layout while satisfying aesthetic and functional constraints. Early works (DeVries et al., 2021; Bautista et al., 2022; Chen et al., 2023; Zhang et al., 2024b) employed generative models to learn holistic 3D scene distributions, e.g., GAN-based unbounded natural scene generation (Liu et al., 2021; Li et al., 2022) or semantic map to radiance field translation (Hao et al., 2021). Recent diffusion-based methods (Fridman et al., 2023; Yu et al., 2024; Höllein et al., 2023; Zhang et al., 2024a; Li et al., 2024) predict 2D content and lift it to 3D via depth estimation, with extensions to panoramic 3D reconstruction (Zhou et al., 2025a). However, these approaches often produce monolithic scene representations, limiting object-level control and fine-grained editability.

Compositional and LLM-guided scene generation has gained traction (Zhai et al., 2023; Epstein et al., 2024b; Paschalidou et al., 2021; Po & Wetzstein, 2024; Gao et al., 2024; Yang et al., 2024b). Layout priors or scene graphs are used to guide generation, and ACDC (Dai et al., 2024) constructs diverse "digital cousin" environments for sim-to-real robustness. More recent methods focus on multimodal 3D world generation, including HOLODECK 2.0 (Bian et al., 2025), HunyuanWorld 1.0 (Team et al., 2025), EmbodiedGen (Wang et al., 2025), SynCity (Engstler et al., 2025), and Schwarz et al. (Schwarz et al., 2025), which explore generation from text, images, or mixed inputs. While these methods advance 3D world generation, most rely on pre-defined assets, provide limited object-level control, or do not maintain dynamic internal representations for agentic reasoning. In contrast, our framework constructs a structured, continually updatable *spatial context*, enabling VLMs to act agentically and reason over both object-level and environment-level constraints.

**Layout Generation.** Accurate object placement is central to compositional scene synthesis, requiring functional, aesthetic, and ergonomic constraints. Early approaches (Kjølaas, 2000; Coyne & Sproat, 2001; Germer & Schwarz, 2009; Yu et al., 2011) relied on rule-based templates or exemplars, limiting generalization. Data-driven methods improve robustness with sequential models (Wang et al., 2021; Paschalidou et al., 2021; Sun et al., 2025b) or denoising diffusion (Para et al., 2023; Tang et al., 2024). Recent works disentangle layout learning from appearance (Epstein et al., 2024a), use layout guidance for complex 3D generation (Zhou et al., 2024), or employ LLMs for text-driven layouts (Fu et al., 2025; Feng et al., 2024; Yang et al., 2025b). Others refine layouts via differentiable objectives (Sun et al., 2025a) or enforce physical plausibility (Zhou et al., 2025b). Yet, most remain dependent on exemplars, struggle with dynamic intent, and rarely model ergonomic principles, open-vocabulary objects, or higher-order relations such as symmetry or equidistance. Our framework addresses these gaps by introducing a *scene hypergraph* that encodes object–object and object–environment interactions and guides ergonomics-aware layout refinement.

**LLMs for Visual Programming.** Large Language Models (LLMs) have demonstrated impressive zero-shot and few-shot reasoning capabilities (Brown et al., 2020; Ouyang et al., 2022; Achiam et al., 2023; Touvron et al., 2023; Dubey et al., 2024; Team et al., 2023), with recent multimodal extensions (Alayrac et al., 2022; Li et al., 2023; Liu et al., 2023) supporting joint text-image reasoning. Tool-augmented agents further leverage APIs and visual foundation models for complex tasks, including visual code synthesis (Wu et al., 2023; Gupta & Kembhavi, 2023; Surís et al., 2023) and multimodal generation/editing (Sharma et al., 2024; Lian et al., 2024; Wu et al., 2024; Feng et al., 2024; Wang et al., 2024; Yang et al., 2024a). SceneCraft (Hu et al., 2024) employs an LLM agent to translate text prompts into 3D scenes via Blender scripting, but lacks explicit spatial grounding and struggles with complex scenes, ergonomic constraints, and open-vocabulary objects. By contrast, our work injects a structured *spatial context* into VLMs, enabling dynamic, geometry-aware internal representations, agentic reasoning, and fine-grained control over 3D scene generation.

**Summary of Differences.** In summary, our work differs from prior 3D scene generation and layout methods in three major ways: (1) we construct a structured, continually updatable *spatial context* that integrates multimodal inputs and partial reconstruction constraints, (2) we explicitly model both object–object and object–environment interactions through a scene hypergraph to guide ergonomics-aware layout and environment setup, and (3) we enable agentic 3D scene generation with fine-grained control over geometry, placement, and appearance, supported by auto-verification, which together provide a level of flexibility, generalization, and semantic alignment not achieved by previous approaches.

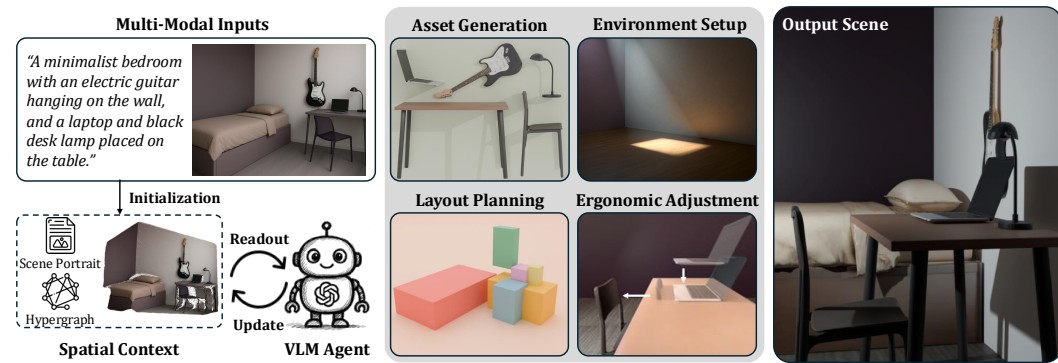

Figure 2: **Left: Spatial Context.** Constructed from multimodal inputs, it consists of a *scene portrait* (semantic blueprints of layout, objects, environment) and a *scene hypergraph* (object–object and object–environment relations). **Right: Agentic Scene Generation.** Injected with this context, the VLM performs *asset synthesis*, *environment setup*, *layout planning*, and *ergonomic adjustment*, supported by an *auto-verification mechanism* that enforces semantic and spatial fidelity.

## 3 METHOD

Our framework is organized into three key components. First, we build a *spatial context* that unifies semantic blueprints and relational constraints (Section 3.1). Second, we describe how the VLM performs *agentic scene generation*, combining asset synthesis and layout optimization (Section 3.2). Finally, we introduce an *auto-verification mechanism* that continuously monitors and corrects the scene to ensure semantic and spatial fidelity (Section 3.3).

### 3.1 SPATIAL CONTEXT CONSTRUCTION

The spatial context serves as a structured, dynamic working memory for the VLM, integrating multimodal input into semantic, geometric, and relational representations that guide agentic 3D scene generation. It unifies scene-level intent and object-level constraints with a relational graph of the environment, and is formally defined as $C = (S, G)$, where $S$ is the *scene portrait* and $G$ is the *scene hypergraph*.

**Scene Portrait.** The VLM constructs a multimodal scene portrait $S$, a high-level structured representation of the scene. The portrait is a *threefold representation* that integrates:

- *Portrait Text.* A structured summary that concisely conveys the overall scene content, style, and atmosphere, describes the spatial layout across foreground, midground, and background regions, and specifies core objects together with their appearance and semantic attributes.

- *Portrait Image.* Either user-provided or synthesized from the textual description when absent, serving as a visual reference to the scene content.

- *Portrait Geometry.* A geometric grounding of the scene, initially generated as a semantically labeled point cloud via Fast3R (Yang et al., 2025a), with each point

$$(\mathbf{x}_i, \mathbf{c}_i, l_i),$$

where $\mathbf{x}_i \in \mathbb{R}^3$ is the 3D coordinate, $\mathbf{c}_i \in \mathbb{R}^3$ the RGB color, and $l_i \in \mathbb{N}$ the semantic label from Grounded-SAM (Ren et al., 2024). For multi-view inputs, detections are merged by spatial overlap and semantic similarity. The point cloud is iteratively refined, while an object-level mesh repository is maintained to improve geometric fidelity and support reliable view interpolation.

**Scene Hypergraph.** To model spatial relationships among objects and environment components, the VLM constructs a *scene hypergraph* $G = (V, E)$ from the object instances and their axis-aligned bounding boxes derived from the scene portrait. Nodes $V$ represent objects, with *special nodes for environment components* such as ground, walls, water ponds, or terrain. Hyper-edges $E$ encode unary, binary, and higher-order relations, including clearance, contact, alignment, symmetry, and equidistance. This hypergraph enables the VLM to reason over object–object and object–environment interactions, supporting layout planning, ergonomic adjustment, and environ-

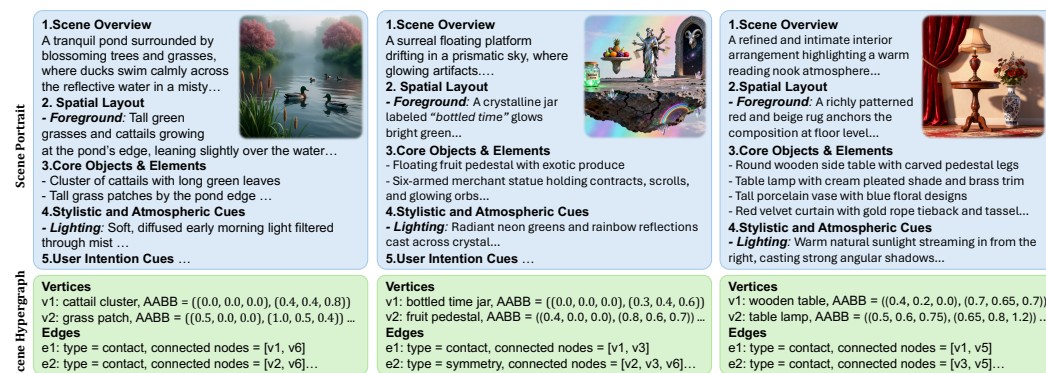

Figure 3: **Spatial Context Overview.** The spatial context unifies semantic, geometric, and relational information to guide agentic 3D scene generation. The **scene portrait** includes a high-level scene overview, spatial layout descriptions (foreground, midground, background), core objects and elements with detailed sub-descriptions, environment setup cues (stylistic and atmospheric), and a semantically labeled point cloud providing geometry. The **scene hypergraph** encodes unary, binary, and higher-order spatial relations among objects and environment components, supporting layout refinement, ergonomic adjustment, and environment construction. Together, these components form a structured, dynamic representation that the VLM reads and updates iteratively during generation.

ment setup. Together with the scene portrait, it forms a structured, dynamic representation of the scene that the VLM iteratively references during the agentic 3D generation process.

## 3.2 AGENTIC SCENE GENERATION

Given the spatial context $C = (S, G)$, our framework performs agentic scene generation, where the VLM synthesizes 3D assets, aligns them with geometric constraints, and refines their placement using relational reasoning.

**Asset generation.** Each core object in the scene portrait is associated with textual sub-descriptions and, when available, semantically labeled point cloud segments extracted using Fast3R. These point clouds are often incomplete due to occlusion or limited viewpoints. To restore missing geometry, we employ a lightweight completion module adapted from Point-M2AE Zhang et al. (2022). The module is trained by randomly masking parts of clean single-object point clouds and reconstructing the full shapes.

For each object instance $v \in V$, we first assess whether the extracted segment $P_v \subset S$ is sufficiently complete. If incomplete, the restoration module produces a densified version $\hat{P}_v$, and the portrait is updated by replacing $P_v$ with $\hat{P}_v$. The restored point cloud is then projected into a canonical front-view rendering and combined with the corresponding textual sub-description from $S$ to guide a 3D asset generator, which synthesizes a textured mesh. The system does not require separate user-provided object-level prompts (e.g., "an electric guitar"). The VLM agent automatically derives object-level descriptions by interpreting the main prompt together with semantic cues in the scene portrait and geometric cues from the canonical front-view rendering of the restored point cloud. This process yields a refined textual sub-description that specifies appearance, style, and functional attributes. The resulting VLM-generated **text prompt**, paired with the **canonical front-view image**, forms the multimodal input to the external 3D asset generator (Meshy API). When users optionally supply more detailed object-level descriptions, the system incorporates them and can further improve asset fidelity.

We maintain a *mesh repository* that stores previously generated assets. When an object reappears in subsequent iterations, the system first retrieves its mesh from the repository. A consistency check compares the mesh geometry against the updated point cloud using the average distance between mesh vertices and nearest point cloud samples. If the discrepancy is below a threshold, the stored mesh is reused directly; otherwise, the mesh is regenerated using the restoration and synthe-

sis pipeline. This strategy balances efficiency and adaptability, ensuring accurate geometry while avoiding unnecessary recomputation.

**Coarse alignment via point cloud fitting.** The generated mesh is placed into the scene by aligning it with its restored point cloud. Let $M_v = \{\mathbf{m}_i\}$ denote mesh vertices and $P_v = \{\mathbf{p}_j\}$ the point segment. We estimate a similarity transformation, scale $s$, rotation $R$, and translation $\mathbf{t}$, by solving

$$(s^*, R^*, \mathbf{t}^*) = \arg \min_{s,R,\mathbf{t}} \sum_i \|sR\mathbf{m}_i + \mathbf{t} - \mathrm{NN}_{P_v}(sR\mathbf{m}_i + \mathbf{t})\|^2, \tag{1}$$

where $\mathrm{NN}_{P_v}(\cdot)$ is the nearest neighbor in $P_v$. Initialization is performed by aligning centroids and OBB axes, followed by ICP refinement. The optimized transformation is recorded in the portrait, ensuring consistency between reconstructed meshes and their geometric constraints.

**Hypergraph-based ergonomic adjustment.** While coarse alignment ensures global consistency with point cloud observations, structural issues such as inter-object penetration, detachment, or misalignment with ergonomic expectations may remain. To resolve these, the VLM performs a joint optimization over object poses guided by the **scene hypergraph** $G = (V, E)$. Nodes represent objects and special environment components (e.g., ground, water pond), while hyperedges encode unary, binary, and higher-order relations such as clearance, contact, alignment, equidistance, and symmetry. We optimize object transformations $\{R_v, \mathbf{t}_v\}_{v \in V}$ to satisfy soft spatial constraints:

$$\min_{\{R_v, \mathbf{t}_v\}_{v \in V}} \sum_{e \in E} \lambda_{r_e} \cdot L_{r_e}(\{R_v, \mathbf{t}_v\}_{v \in e}), \tag{2}$$

where $L_{r_e}$ is a relation-specific loss and $\lambda_{r_e}$ its weight.

*Relation-specific loss.* As a representative case, the contact loss encourages physical contact between two objects $v_i$ and $v_j$. Let $M_{v_i}$ and $M_{v_j}$ be sampled surface points. After transformation, points are $\tilde{\mathbf{p}} = R_{v_i}\mathbf{p} + \mathbf{t}_{v_i}$ and $\tilde{\mathbf{q}} = R_{v_j}\mathbf{q} + \mathbf{t}_{v_j}$. The loss is:

$$L_{\text{contact}} = \left[\min_{\mathbf{p}, \mathbf{q}} \|\tilde{\mathbf{p}} - \tilde{\mathbf{q}}\| - \epsilon\right]_+^2, \tag{3}$$

where $[\cdot]_+ = \max(0, \cdot)$ and $\epsilon$ is a soft margin. Other relation losses (alignment, clearance, symmetry, equidistance) are defined analogously (see supplementary material).

### 3.3 AUTO-VERIFICATION MECHANISM

To ensure that the generated 3D scene satisfies the constraints specified in the spatial context, we introduce an **auto-verification agent**. This agent continuously monitors and validates the scene at the object and environment levels, enabling reliable integration of VLM-generated content with structured 3D guidance.

**Context Readout.** The VLM reads from the spatial context $C = (S, G)$ to guide verification. The **scene portrait**, comprising structured text and images, provides semantic and stylistic reference, which can be directly interpreted by the VLM. The **scene hypergraph**, expressed in textual form, encodes relational constraints among objects and environment components, including unary, binary, and higher-order relations.

The **semantically labeled point cloud**, which encodes the 3D geometry of reconstructed scene and objects, is projected into 2D RGB+instance maps from all available input viewpoints. For single-view inputs, additional canonical orthographic projections (top-down, side views) are used. When available, **mesh models retrieved from the repository** are rendered, providing more accurate geometry and appearance cues for verification. These projections preserve sufficient spatial and semantic information for the VLM to assess scene correctness without requiring native 3D processing.

**Context Update.** When the VLM generates or modifies an object $v \in V$, e.g., through asset replacement or geometric adjustment, the corresponding point cloud segment $P_v \subseteq P$ is extracted from the global point cloud $P = \{(\mathbf{x}_i, \mathbf{c}_i, l_i)\}_{i=1}^N$ using the instance labels. After producing the revised segment $\hat{P}_v$, the global point cloud is updated as

$$P \leftarrow (P \setminus P_v) \cup \hat{P}_v.$$

**Verification Process.** The auto-verification agent checks that:

*Sherlock Holmes's 221B Baker Street apartment.*

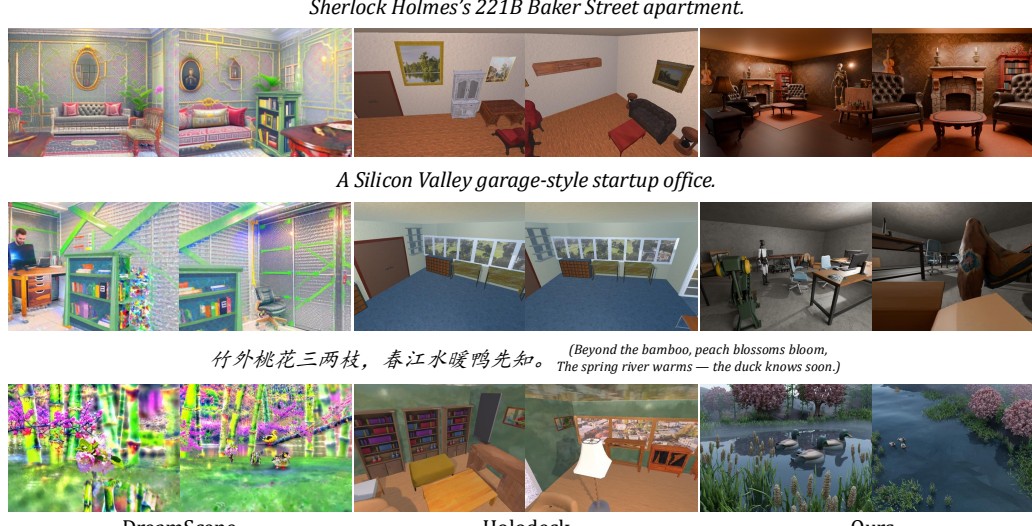

*A Silicon Valley garage-style startup office.*

竹外桃花三两枝，春江水暖鸭先知。 *(Beyond the bamboo, peach blossoms bloom, The spring river warms — the duck knows soon.)*

DreamScene        Holodeck        Ours

Figure 4: **Qualitative comparison for text-based 3D scene generation.** Our method produces more coherent, stylistically aligned, and visually plausible scenes compared to DreamScene (Li et al., 2024) and Holodeck (Yang et al., 2024b).

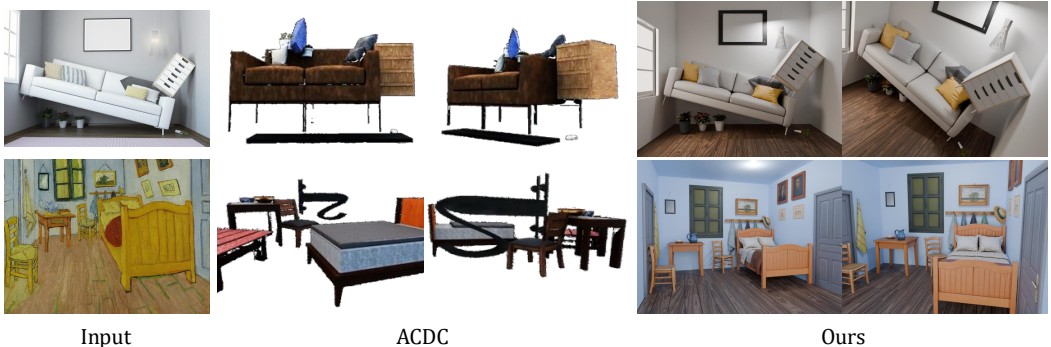

Input        ACDC        Ours

Figure 5: **Qualitative comparison for image-based 3D scene generation.**

- Object placements and scales satisfy spatial layout and hypergraph constraints (e.g., contact, alignment, symmetry).
- Partial reconstruction constraints derived from the scene portrait, including textual, visual, or point cloud cues, are preserved.
- Environment components (e.g., ground, water bodies) remain consistent with object–environment interactions specified in the hypergraph.

If any discrepancies are detected, the agent flags the object or scene region for refinement. The VLM then re-generates or adjusts the relevant content, and the context is updated accordingly. This closed-loop mechanism ensures that the scene remains semantically coherent, geometrically accurate, and faithful to user-specified constraints, while allowing iterative updates during agentic 3D scene generation.

## 4 EXPERIMENTS

We evaluate our proposed framework for 3D scene generation across diverse challenging scenarios, and include comparisons with SOTA baselines and ablation studies to validate the effectiveness of

Table 1: **Quantitative comparison on 3D scene generation.** Our method achieves the best performance across consistency (CLIP/BLIP), image fidelity (LPIPS), aesthetics (AQ), and functionality (FP).

| Method | CLIP (↑) | BLIP (↑) | LPIPS (↓) | AQ (4o/User) (↑) | FP (4o/User) (↑) |
|---|---|---|---|---|---|
| Holodeck | 0.274 | 0.461 | - | 3.00 / 3.25 | 3.00 / 2.69 |
| DreamScene | 0.219 | 0.509 | - | 4.00 / 2.75 | 4.00 / 2.75 |
| ACDC | - | - | 0.760 | 2.00 / 2.94 | 2.00 / 3.31 |
| **Ours** | **0.385** | **0.737** | **0.571** | **1.00 / 1.06** | **1.00 / 1.19** |

key components. We further demonstrate the capabilities of the spatially contextualized VLM in performing downstream spatially grounded tasks. For additional results and implementation details, please refer to our *supplementary material and accompanying video*. All quantitative evaluations follow a consistent setup: we use 30 prompts in total (10 text-only and 10 single-image). For each generated scene, we render five RGB views at a resolution of $960 \times 540$ using randomly sampled camera poses, resampling invalid viewpoints. These rendered views serve as the basis for all quantitative metrics reported in Table 1.

**Implementation details.** We adopt GPT-4o (Achiam et al., 2023) as the VLM, integrating the spatial context and acting as the agent throughout the 3D scene generation pipeline. *Prompts used to construct the spatial context are provided in the appendix.* Our geometric restoration module is trained on point maps estimated by Fast3R (Yang et al., 2025a) using CO3D (Reizenstein et al., 2021) training images, converging in about 3 hours on an NVIDIA A100 GPU. Asset generation uses the Meshy API[1] for image-to-3D synthesis, and layout planning with ergonomic adjustment is implemented via PyTorch optimization. All final 3D scenes are rendered with Blender Cycles for photorealistic results and accurate lighting and materials.

**Metrics.** To evaluate semantic alignment with input prompts, we render images from the generated 3D scenes and compute text–image similarity using *CLIP* (Radford et al., 2021) and *BLIP* (Li et al., 2023), and image–image similarity using *LPIPS* (AlexNet) (Zhang et al., 2018). CLIP and BLIP scores are averaged over the five rendered views, and LPIPS is computed between the input image and the best-matching rendered view in image-conditioned experiments. To assess *aesthetic quality* (realism and visual appeal) and *functional plausibility* (ergonomic adherence), we collect human and GPT-4o ratings and report relative rankings across methods. In Table 1, each method is ranked by averaged ordinal scores over a benchmark set, with lower ranks indicating better performance. Human ratings come from a study with 16 participants. These values represent averaged ordinal ranks, where each rater ranks all methods per scene (1 = best). The reported numbers are normalized mean ranks across scenes rather than absolute scores, so lower values indicate better performance and are directly comparable across methods.

## 4.1 COMPARISON

**Text-conditioned generation.** We compare our framework against two recent text-to-3D methods: Holodeck (Yang et al., 2024b) and DreamScene (Li et al., 2024). As shown in Figure 4, our method produces scenes that more faithfully preserve semantic alignment, spatial structure, and stylistic intent. For example, in the *Holmes apartment* case, our result better captures the Victorian mood in layout and furniture arrangement, while others exhibit geometric artifacts or overlook contextual cues. Quantitatively, our method achieves the highest CLIP and BLIP scores, reflecting superior consistency with input prompts. It also ranks best in aesthetic quality (AQ) and functional plausibility (FP), based on both GPT-4o and user evaluations.

**Image-conditioned generation.** Figure 5 shows a comparison with ACDC (Dai et al., 2024), a recent method for real-to-sim scene construction. Our system more accurately reconstructs spatial layouts and scene composition, such as the tilted sofa in a living room or stylistic integrity in Van Gogh's *Bedroom in Arles*. In Table 1, we report the best image-image similarity score, indicating higher visual fidelity to the input images. This advantage stems from our structured spatial context, which preserves geometric details and enables adaptive reconstruction.

---

[1]https://www.meshy.ai/api

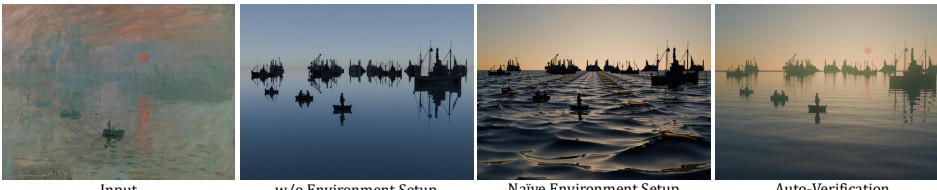

| Input | w/o Environment Setup | Naïve Environment Setup | Auto-Verification |

Figure 6: **Ablation on environment setup**. Without structured setup, scenes lack realistic lighting and environmental elements. Naïve modifiers yield low-fidelity results, while our auto-verified setup produces coherent, atmospheric environments aligned with spatial context.

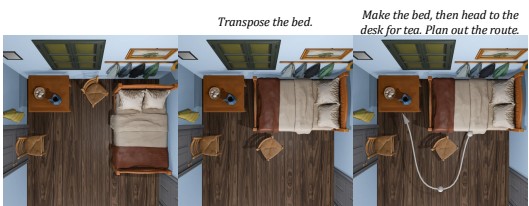

*Transpose the bed.*   *Make the bed, then head to the desk for tea. Plan out the route.*

Figure 7: **Scene editing and spatial reasoning.** Our method enables downstream spatial tasks such as furniture manipulation and obstacle-aware path planning, by reasoning over the spatial context.

**Image set as input.** Unlike prior methods, which are typically restricted to single-view inputs, our framework naturally accommodates unstructured and unposed image collections. As illustrated in Figure 8, our system consolidates geometric cues from diverse viewpoints into a coherent 3D layout. This ability stems from the VLM's integration with our spatial context, which provides a flexible representation for resolving spatial correspondences across views.

### 4.2 ABLATION STUDY

**Core building components.** Table 2 reports both semantic metrics (CLIP/BLIP) and simple geometric validity metrics (collision rate and support-violation rate, where collisions count object–object interpenetration and support violations indicate objects whose bottom surfaces are not in valid physical contact with the floor or another supporting object) , providing a high-level validation of our design. Removing any major component leads to consistent drops in semantic alignment and increases in geometric errors. This pattern reflects the complementary roles of multimodal grounding, relational structure, and iterative verification in stabilizing the agentic generation process. The full model integrates these signals most effectively, supporting the necessity of the complete spatial context.

**Environment Setup.** We evaluate the importance of environment setup and auto-verification. As shown in Figure 6, without this module, key visual elements, such as sky, sunlight, or water, are missing or unnatural. A naïve setup with basic modifiers adds some structure, but often lacks realism—waves may appear flat or physically implausible. In contrast, our auto-verified environment setup enhances scene realism and atmosphere by aligning with the spatial context and refining visual fidelity through iterative code correction. We evaluate the importance of environment setup and auto-verification. As shown in Figure 6, without this module, key visual elements, such as sky, sunlight, or water, are missing or unnatural. A naïve setup with basic modifiers adds some structure, but often lacks realism—waves may appear flat or physically implausible. In contrast, our auto-verified environment setup enhances scene realism and atmosphere by aligning with the spatial context and refining visual fidelity through iterative code correction.

**Layout Planning.** We assess layout planning by replacing our method with ATISS (Paschalidou et al., 2021) and LayoutGPT (Feng et al., 2024). As shown in Figure 9, these alternatives often introduce scale or placement errors (e.g., floating lamps, misaligned furniture), whereas our method yields more structurally accurate and semantically coherent layouts. Removing ergonomic adjust-

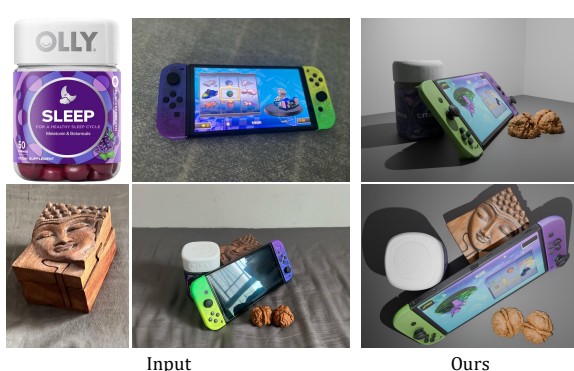

Figure 8: **Results from multi-view observations.** Our method synthesizes consistent scenes from unposed, unstructured image collections.

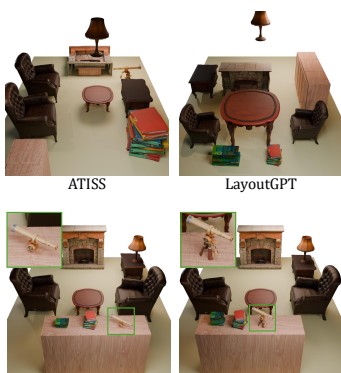

Figure 9: **Ablation on layout planning and ergonomic adjustment.**

Table 2: **Ablation on core building components.**

| Variant | CLIP ↑ | BLIP ↑ | Collisions (%) ↓ | Support viol. (%) ↓ |
|---|---|---|---|---|
| w/o hypergraph | 0.361 | 0.674 | 36.0 | 16.0 |
| w/o portrait | 0.288 | 0.465 | 12.5 | 15.6 |
| w/o auto-verification | 0.313 | 0.605 | 28.1 | 21.1 |
| **Ours** | **0.385** | **0.737** | **5.8** | **3.8** |

ment results in object misalignment and interpenetration, leading to degraded visual aesthetics and functional plausibility. These findings highlight the necessity of our ergonomic refinement step for ensuring realistic and usable 3D scenes.

### 4.3 SPATIALLY GROUNDED DOWNSTREAM TASKS

Our framework supports downstream spatial tasks such as object manipulation and navigation planning. As shown in Figure 7, the VLM can follow high-level instructions, e.g., relocating furniture or planning a route. It can generate a collision-free path from the bed to the desk without explicit labels or obstacle maps by implicitly understanding the spatial layout and avoiding objects like the bedside chair. This is enabled by our structured spatial context, which encodes object geometry and relations and is dynamically updated after editing, allowing the VLM to extract feasible trajectories from the modified scene.

## 5 CONCLUSION

We present **Spatially Contextualized VLMs**, an agentic framework for high-fidelity 3D scene generation guided by multimodal spatial context. By combining a structured **scene portrait** with a **scene hypergraph**, our method unifies semantic intent, geometric constraints, and relational reasoning, enabling iterative object reconstruction and layout refinement. Key components include partial point cloud-guided geometric reconstruction, hypergraph-based ergonomic adjustment, and a closed-loop auto-verification agent ensuring semantic and physical consistency. This framework enables the creation of coherent, physically plausible, and interactive 3D environments from sparse or heterogeneous inputs, providing a foundation for future research in VLM-driven 3D synthesis and embodied AI.

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

APPENDIX

## A  LIMITATIONS AND FUTURE WORK

While our framework demonstrates strong generalization and performance, several limitations remain. First, when the number of object instances is large or includes extremely small objects, spatial context construction may miss instances or introduce noise, potentially affecting layout quality and scene completeness. Second, in the multi-image setting, performance heavily relies on the geometric foundation model used to estimate depth and structure—failure cases in depth prediction can lead to misalignment in the resulting scene. Finally, our current scene hypergraph models unary, binary, and ternary spatial relations; extending this structure to support richer or learned higher-order relations could further enhance ergonomic reasoning and compositional flexibility. Addressing these challenges offers promising directions for future work.

## B  PROMPT DESIGN FOR SPATIAL CONTEXT INITIALIZATION

---

**Scene Portrait Construction Prompt**

You are a visual and spatial reasoning expert. Your task is to analyze user-provided input—either text (e.g., a poem, abstract description, or narrative) or an image—and return a well-structured Scene Portrait. This portrait functions as an implementation-ready blueprint for 3D scene generation, immersive rendering, or environment design.

Input - A text description or reference images or their combination

Output - Scene Portrait (Structured) Please return the scene portrait using the following format:

1. Scene Overview

A concise, one-sentence summary of the scene's overall atmosphere, setting, and intent.

2. Spatial Layout

Describe the division of space (foreground, midground, background). Include the positioning of key objects or actors, and any notable spatial relationships or focal points.

3. Core Objects & Elements

List the concrete, self-contained assets that physically compose the scene. Each item must be individually identifiable and renderable. Avoid diffuse effects or global states as standalone elements.

Good asset examples: - "Wooden bookshelf filled with leather-bound books" - "A vintage painting of a sailing ship above the fireplace" - "Glass coffee table with a silver tea set"

Bad asset examples (do not list): - "Walls", "Floors", "Foggy atmosphere", "Golden sunlight", "Dense mist", "Interior"

4. Stylistic and Atmospheric Cues

Describe the lighting, color palette, material textures, era, and cultural or thematic styling. This section defines the look and feel of the scene and complements the concrete assets above.

Indoor scenes: - Always specify lighting conditions (e.g., warm lamplight, cool overhead daylight) - Describe architectural elements like wall material, flooring, and ceiling structure

Outdoor scenes: - Always specify weather and sky conditions (e.g., foggy, overcast, golden hour) - Include ground material (e.g., stone path, muddy field) - Mention terrain features or vegetation

5. User Intention Cues (Optional) If any emotional tone, narrative theme, or symbolic layer is implied in the input, capture it here. Examples: "The scene evokes nostalgia and quiet reflection", "Suggests confrontation between civilization and nature"

Reference Example

User Input: "Oval Office, White House"

Scene Portrait

1. Scene Overview

A stately, formal executive office representing U.S. presidential authority, diplomacy, and legacy.

2. Spatial Layout

Foreground: Two beige-upholstered armchairs face a glass coffee table set on the presidential seal rug. Center: The Resolute Desk, with the presidential chair behind it, faces the room entrance. Background: Three tall curtained windows frame the back wall, flanked by symmetrical bookshelves and decorative columns.

3. Core Objects & Elements

- The Resolute Desk with leather blotter, pen set, and phone - Two beige armchairs with dark wood trim - Round glass coffee table - A large American flag and presidential seal rug - Framed portrait of George Washington above the fireplace - Twin lamps on the bookshelf - Floor globe near the desk

4. Stylistic and Atmospheric Cues

- Lighting: Soft daylight entering through sheer curtains, enhanced by two warm-toned lamps - Color palette: Navy blue, cream, gold - Materials: Mahogany wood, brass accents, polished glass - Era: Mid-20th to modern - Style: Neoclassical with modern diplomatic elegance

5. User Intention Cues

The arrangement communicates authority, control, and ceremonial readiness for public-facing leadership.

---

---

**Scene Hypergraph Construction Prompt**

You are a spatial reasoning module responsible for constructing a scene hypergraph from a set of 3D object instances. Your goal is to infer a hypergraph representation that captures spatial and ergonomic relationships among these objects.
Input: A list of object instances (vertices), where each instance includes:
- Class label (e.g., "chair", "table", "monitor")
- Axis-aligned bounding box (AABB), specified by:
- min corner: (x_min, y_min, z_min)
- max corner: (x_max, y_max, z_max)
Output: A scene hypergraph consisting of:
- Vertices: all input object instances, indexed as v1, v2, ..., vn
- Edges: a set of spatial relationships, each defined by: - Type: clearance (unary), contact or alignment (binary), equidistance or symmetry (ternary) - Connected nodes: list of node IDs involved
Example
Input:
v1: class = "chair", AABB = ((0.75, 1.25, 0), (1.25, 1.75, 1.0))
v2: class = "chair", AABB = ((1.75, 1.25, 0), (2.25, 1.75, 1.0))
v3: class = "dining table", AABB = ((0.75, 0.60, 0), (2.25, 1.40, 0.75))
v4: class = "fan", AABB = ((2.85, 1.85, 0), (3.15, 2.15, 1.2))
Output: e1: type = contact, connected nodes = [v1, v3]
e2: type = contact, connected nodes = [v2, v3]
e3: type = alignment, connected nodes = [v1, v2]
e4: type = symmetry, connected nodes = [v1, v2, v3]
e5: type = clearance, connected nodes = [v4]

## C   Ergonomic Adjustment: Relation-Specific Constraints

In this section, we detail the definitions of other relation-specific loss functions used in our ergonomic adjustment module, as referenced in Section 4.4. While the main text introduces the contact constraint, our scene hypergraph formulation supports a richer set of spatial relations—including unary (e.g., clearance), binary (e.g., alignment), and ternary (e.g., symmetry, equidistance). Each is encoded as a soft differentiable loss to guide physically plausible and semantically meaningful spatial arrangements. Below, we present the mathematical formulation and intuition behind each additional constraint type.

**Clearance.** To prevent spatial crowding and ensure functional space around objects, we introduce a unary clearance constraint that enforces a minimum separation between each object and all others in the scene. Let $\mathbf{o}_v$ denote the center of the axis-aligned bounding box (AABB) of object $v$ in its local frame. After transformation, its world-space position is $\tilde{\mathbf{o}}_v = R_v \mathbf{o}_v + \mathbf{t}_v$. For each object $v \in V$, the clearance loss is defined as:

$$L_{\text{clearance}}(R_v, \mathbf{t}_v) = \sum_{\substack{v' \in V \\ v' \neq v}} \left[ d_{\min}(v) - \| \tilde{\mathbf{o}}_v - \tilde{\mathbf{o}}_{v'} \| \right]_+^2 , \tag{4}$$

where $d_{\min}(v)$ is a VLM-determined minimum clearance radius for object $v$, typically computed from its bounding box size or semantic role, and $[\cdot]_+ = \max(0, \cdot)$ denotes the hinge function.

**Alignment.** To promote symmetric or functional alignment between two objects $v_i$ and $v_j$—such as centering a chair relative to a desk—we impose a soft constraint that minimizes their displacement along contextually relevant axes. Let $\mathbf{o}_{v_i}$ and $\mathbf{o}_{v_j}$ denote the centers of the axis-aligned bounding boxes (AABBs) of the respective meshes. After applying transformations, the world-space centers become $\tilde{\mathbf{o}}_{v_i} = R_{v_i} \mathbf{o}_{v_i} + \mathbf{t}_{v_i}$ and $\tilde{\mathbf{o}}_{v_j} = R_{v_j} \mathbf{o}_{v_j} + \mathbf{t}_{v_j}$. The alignment loss is defined as:

$$L_{\text{align}}(R_{v_i}, \mathbf{t}_{v_i}, R_{v_j}, \mathbf{t}_{v_j}) = \left\| \mathbf{A}_{r_{ij}} \left( \tilde{\mathbf{o}}_{v_i} - \tilde{\mathbf{o}}_{v_j} \right) \right\|^2 , \tag{5}$$

where $\mathbf{A}_{r_{ij}} \in \mathbb{R}^{d \times 3}$ is a projection matrix that selects the axis or axes relevant to the alignment relation $r_{ij}$. This encourages alignment along those axes while allowing flexibility in other directions.

**Symmetry.** To encourage symmetric spatial arrangements, we introduce a ternary symmetry constraint. It ensures that two objects $v_i$ and $v_j$ are symmetrically positioned with respect to a reference object $v_k$ along a contextually relevant axis. The axis of symmetry—typically one of the global $x$, $y$, or $z$ axes—is determined by the VLM based on semantic roles or scene structure. Let $\tilde{\mathbf{o}}_v = R_v \mathbf{o}_v + \mathbf{t}_v$ denote the transformed AABB center of object $v \in \{v_i, v_j, v_k\}$. Let $\mathbf{A}_r \in \mathbb{R}^{1 \times 3}$

be the axis selector vector corresponding to the symmetry relation $r \in \{x, y, z\}$, e.g., $\mathbf{A}_x = [1, 0, 0]$. The symmetry loss is defined as:

$$L_{\text{symmetry}} = \left\| \mathbf{A}_r \left( \frac{\tilde{\mathbf{o}}_{v_i} + \tilde{\mathbf{o}}_{v_j}}{2} - \tilde{\mathbf{o}}_{v_k} \right) \right\|^2,$$ (6)

which penalizes deviation of the midpoint between $v_i$ and $v_j$ from the center of $v_k$ along the symmetry axis.

**Equidistance.** To enforce symmetric spacing, we introduce an equidistance constraint where two objects $v_i$ and $v_j$ are encouraged to maintain equal distance from a reference object $v_k$ along a specified axis. Let $\tilde{\mathbf{o}}_v = R_v \mathbf{o}_v + \mathbf{t}_v$ denote the transformed AABB center for each $v \in \{v_i, v_j, v_k\}$, and let $\mathbf{a} \in \mathbb{R}^3$ be a unit vector representing the axis of comparison. The equidistance loss is defined as:

$$L_{\text{equi}} = \left\| \mathbf{a}^\top (\tilde{\mathbf{o}}_{v_i} - \tilde{\mathbf{o}}_{v_k}) - \mathbf{a}^\top (\tilde{\mathbf{o}}_{v_j} - \tilde{\mathbf{o}}_{v_k}) \right\|^2.$$ (7)

This loss encourages $v_i$ and $v_j$ to be placed symmetrically with respect to $v_k$ along axis $\mathbf{a}$.

# D LLM USAGE DECLARATIONS

We declare that Large Language Models (LLMs) were used in a limited capacity during the preparation of this manuscript. Specifically, LLMs were employed for grammar checking, word choice refinement, and typo correction. All core technical contributions, experimental design, analysis, and conclusions are entirely our own. The use of LLMs did not influence the scientific methodology, result interpretation, or theoretical contributions of this research.

- v8: pink blossoming tree (right),
AABB =((3.5, 1.0, 0.0), (4.5, 2.0, 2.5))
+ v8: pink blossoming tree (right),
AABB =((-0.5, 1.0, 0.0), (0.5, 2.0, 2.5))

+ e8: type = clearance,
connected nodes = [v9]

+ *Weather/Sky*: Misty morning,
overcast with hazy atmosphere

Time

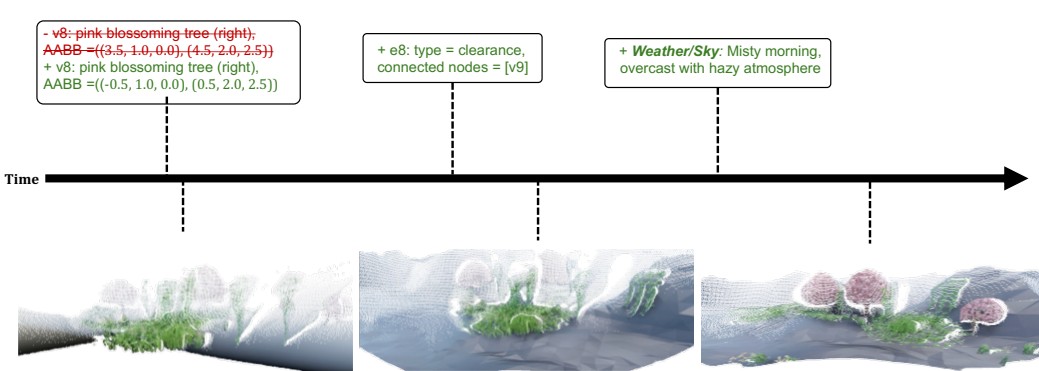

Figure 10: **Demonstration of scene context evolution as the generation proceeds.**

*A dystopian set design reminiscent of Blade Runner 2049.*

竹外桃花三两枝，
春江水暖鸭先知。

(Beyond the bamboo, peach blossoms bloom,
The spring river warms — the duck knows soon.)

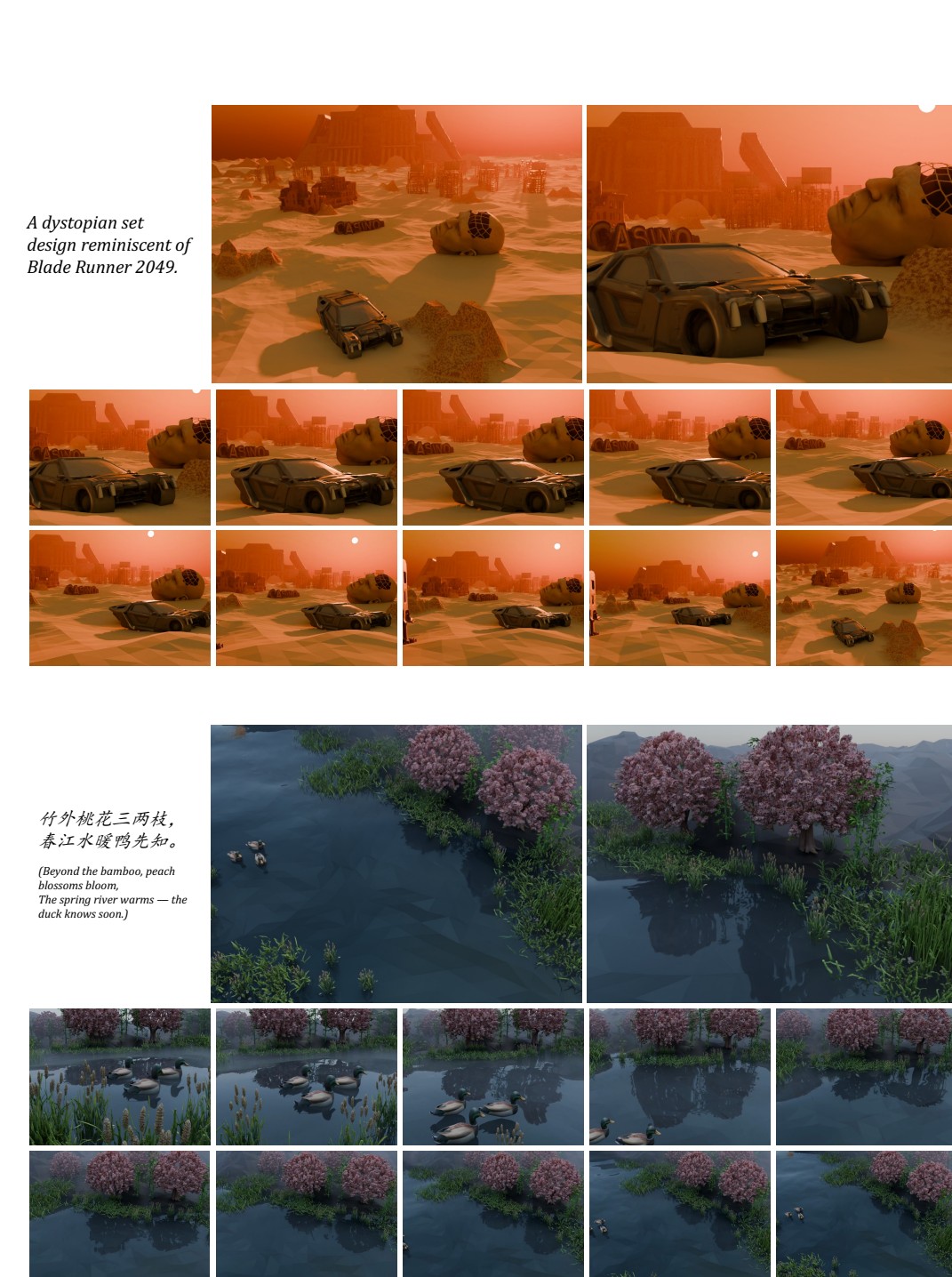

Figure 11: **Additional qualitative results.**

*"A minimalist bedroom with an electric guitar hanging on the wall, and a laptop and black desk lamp placed on the table."*

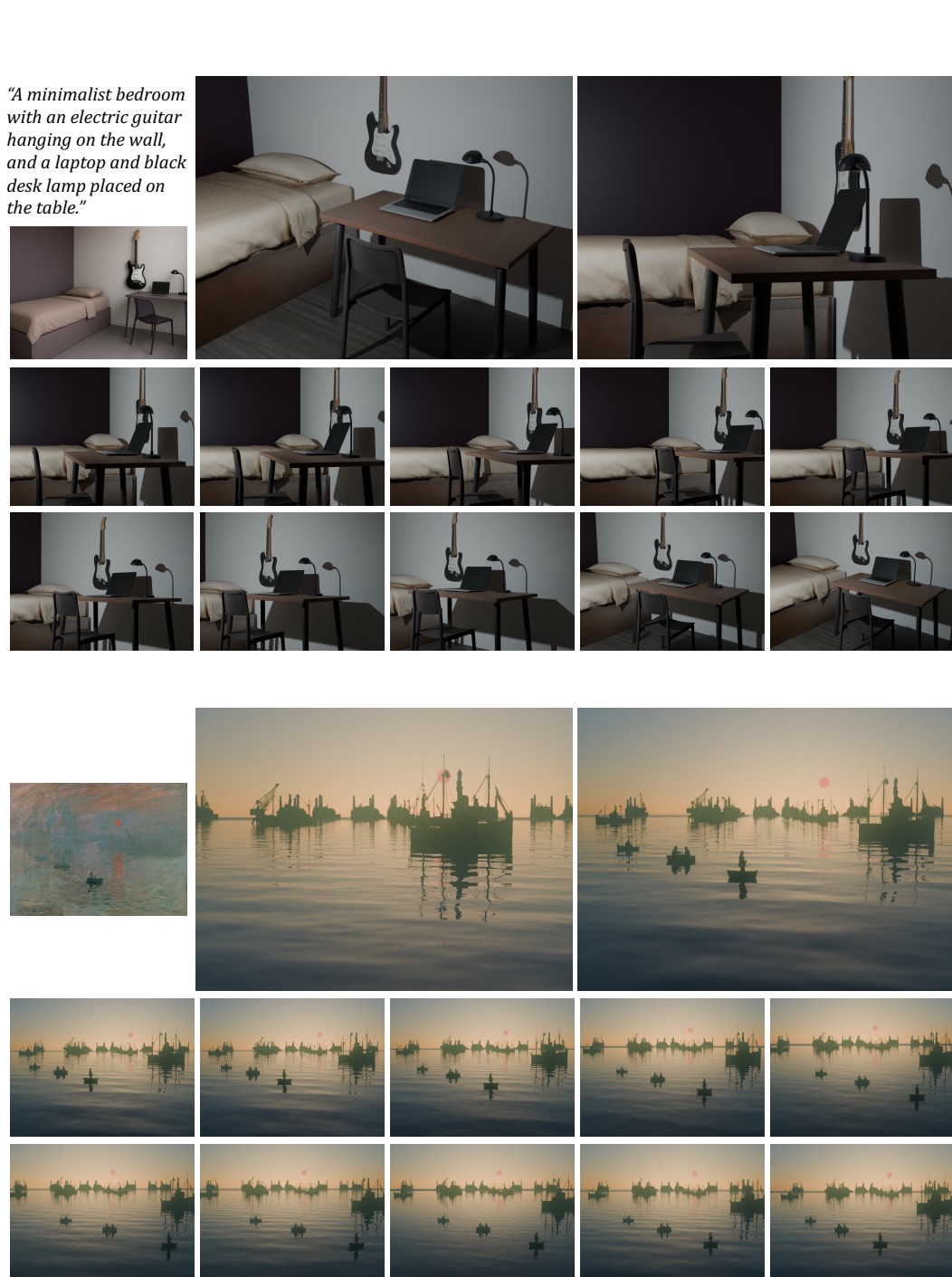

Figure 12: **Additional qualitative results.**

Input Images                                  Generated Scenes

Figure 13: **Additional qualitative results.**

