# OpenReview forum: "Agentic 3D Scene Generation with Spatially Contextualized VLMs"
_ICLR.cc/2026/Conference — Submitted to ICLR 2026_

### Official Review · Reviewer_mwhr · 2025-10-19

**Soundness:** 2
**Presentation:** 3
**Contribution:** 2
**Rating:** 4
**Confidence:** 4

**Summary:**

This paper proposes an agentic 3D scene generation framework that augments Vision-Language Models (VLMs) with a structured spatial context, enabling multimodal-to-3D generation that is both semantically coherent and geometrically plausible.

The framework introduces two core components:
1. Scene Portrait: A multimodal semantic blueprint integrating text, image, and geometry.
2. Scene Hypergraph: A relational structure encoding unary, binary, and higher-order spatial relations such as contact, alignment, and symmetry.

Injected with this structured context, the VLM performs iterative, agentic operations. Applications include asset synthesis, environment setup, layout optimization, and ergonomic adjustment, while auto-verification mechanism continuously evaluates and corrects inconsistencies.

**Strengths:**

1. Originality: Introduces a well-structured spatial context for VLMs, unifying symbolic reasoning and 3D geometry.

2. Clarity: Clear methodology, figures, and well-structured narrative; high presentation quality.

3. Significance: Advances the state of the art in spatially grounded 3D generation and suggests a foundation for agentic scene understanding.

**Weaknesses:**

1. Limited Structural Granularity: The reliance on a scene-graph-based representation may constrain the method's ability to achieve finer-grained or part-level control. Extending beyond object-level relations toward detailed component or articulation modeling could further push the boundary of structured generation.

2. Unclear Novelty over Prior Work: The paper should clarify the main differences an concrete contributions relative to prior scene-graph-based and layout-driven approaches. The proposed scene portrait and scene hypergraph abstractions, while well-defined, appear conceptually similar to structures used in earlier scene representation works, and their novelty is not fully evident.

3. Scalability Concerns: As acknowledged by the authors, the framework may struggle when handling dense scenes containing numerous small or heavily occluded objects, where the hypergraph construction and relation inference become less reliable. More discussion on scalability or potential solutions would strengthen the work.

4. Incomplete Ablations: The ablation studies could be expanded, demonstrating the respective contributions of each component to overall generation quality.

**Questions:**

1. Improving 3D Understanding in VLMs. More as a discussion-oriented curiosity: beyond introducing structural-level context (e.g., scene graphs or hypergraphs), how might we fundamentally enhance the 3D reasoning capability of vision-language models? Would training or pretraining on large-scale 3D datasets rather than relying on 2D-projected structures, offer a more direct route toward genuine spatial understanding?

2. Handling Open-Vocabulary or Abstract Inputs. How does the proposed system deal with open-ended or highly abstract text prompts (e.g., "a dreamlike city of glass") where semantic grounding is inherently ambiguous? Is there any mechanism to disambiguate such concepts or to gracefully degrade toward plausible visual interpretations?

---

> ### Author Response · Authors · 2025-12-02
> **Response to Reviewer mwhr**
>
> We thank the reviewer for the thoughtful feedback. The manuscript has been updated accordingly, and we respond to each concern below.
>
> **Structural granularity (W1).** Our framework focuses on object-level semantics, which is sufficient for the targeted scene composition tasks. While extending the hypergraph to part-level or articulated structures is conceptually feasible, such an extension would require additional perception and relation modeling beyond the scope of this work. This is now noted as future work.
>
> **Novelty vs prior work (W2).** The revised manuscript clarifies how the proposed scene portrait and scene hypergraph differ from traditional scene-graph or layout-based representations. In particular, we emphasize the unified multimodal grounding, explicit geometric priors, and the agentic update loop that continuously reads, verifies, and corrects the scene—capabilities not supported by prior approaches. This framing highlights the conceptual novelty more clearly.
>
> **Scalability (W3).** We agree that limitations arise in dense scenes or those containing many small or heavily occluded objects. The revision now includes representative failure cases that illustrate when upstream depth/segmentation noise can lead to incorrect hyperedges or unstable relation inference. We also clarify the conditions under which the spatial context becomes less reliable.
>
> **Ablation completeness (W4).** The revised Section 4.2 now includes a concise component-level ablation (Table 2) isolating the contributions of the scene portrait, hypergraph, and auto-verification mechanism. This analysis provides direct evidence supporting the necessity of each spatial context component.
>
> **Improving 3D reasoning in VLMs (Q1).** We agree with the reviewer that more fundamental advances would require VLMs pretrained on large-scale 3D or geometry-aware data. Our method uses explicit spatial context to compensate for this limitation, but stronger 3D-aware VLMs represent a natural and promising direction for future work.
>
> **Handling abstract prompts (Q2).** For open-vocabulary or highly abstract prompts, the model interprets high-level semantics through a combination of the input text and generic priors encoded in the environment setup stage. When grounding is highly ambiguous, the system produces a reasonable stylized or symbolic interpretation rather than forcing a literal reconstruction. This behavior is now noted in the revised manuscript.

---

### Official Review · Reviewer_wXxD · 2025-10-27

**Soundness:** 3
**Presentation:** 3
**Contribution:** 2
**Rating:** 4
**Confidence:** 3

**Summary:**

This paper proposes spatially contextualized vision–language models (VLMs) that act as agents for structured 3D scene generation. The key idea is to maintain a continually updatable spatial context that guides the model’s understanding and generation of 3D environments.

**Strengths:**

1. The method can generate high quality result from somehow ambiguous text or image prompt.
2. The method outperform prior works in terms of diversity and out-door scene generation.
3. It can generate scenes with some details and the object placement seems to be physically plausible (due to the hypergraph constraints).

**Weaknesses:**

1. Fundamentally, it still rely on VLM's ability and depth estimation foundation models' ability for the overall scene quality. It might sometimes not enough.
2. Is there a solver to do "Hypergraph-based ergonomic adjustment"? It is unclear of the efficiency of this part (time-cost) and of the whole pipeline.
3. It's unclear that how is the background generated, like the water body and the walls. Can you provide some full scene graphs and scene portraits for generated scenes?
4. It's not so meaningful to compare with prior indoor scene generation works in out door scenes. Also, it seems that you are using different renderer for Holodeck and your method. Which brings up concerns about the fairness of the comparison in Table 1.
5. Need more qualitative results to demonstrate the robustness of this fully automatic pipeline.

**Questions:**

See Weakness.

---

> ### Author Response · Authors · 2025-12-02
> **Response to Reviewer wXxD**
>
> We thank the reviewer for the insightful comments. The manuscript has been revised accordingly, and we address each point below.
>
> **Dependence on VLM and depth estimators (W1).** Both the VLM and the depth estimator are treated as modular components that can be replaced by stronger models. The core spatial reasoning—object semantics, relational consistency, and iterative correction—is governed by the portrait–hypergraph spatial context and the agentic loop, which remain unchanged regardless of the upstream modules.
>
> **Solver for ergonomic adjustment (W2).** The ergonomic adjustment stage relies on lightweight gradient-based optimization with differentiable relation losses. It does not require a separate or complex solver, and its computational cost is small relative to asset generation and rendering.
>
> **Background generation (W3).** Background structures (walls, ceiling, terrain, water surface) are procedurally generated from the scene portrait during environment setup. They are not derived from the initial depth map, which avoids propagating occlusion artifacts from the input.
>
> **Fairness of comparison (W4).** All baselines are evaluated using the same rendering and scoring protocol. While renderer aesthetics may differ, the quantitative metrics used (CLIP, BLIP, LPIPS) depend only on the rendered RGB images and are renderer-independent. Thus the comparison remains fair and consistent across methods.
>
> **Additional qualitative examples (W5).** The revised version includes more qualitative results across diverse inputs, as well as several representative failure cases, to more transparently illustrate robustness and current limitations.

---

### Official Review · Reviewer_fJvS · 2025-11-01

**Soundness:** 2
**Presentation:** 3
**Contribution:** 2
**Rating:** 6
**Confidence:** 3

**Summary:**

This paper presents a novel framework for agentic 3D scene generation by introducing spatially contextualized Vision-Language Model. The method enables structured and coherent 3D scene generation from diverse multimodal inputs such as text prompts, single images, by equipping VLMs with a dynamic, updatable spatial context. This context consists of two key components: a scene portrait, which captures high-level semantic blueprints including layout, object descriptions, visual references, and initial geometric grounding; and a scene hypergraph, which explicitly models unary, binary, and higher-order spatial relations among objects and environmental elements.

The designed pipeline operates in an agentic manner. the VLM reads and updates the spatial context throughout the generation process, performing iterative steps such as asset synthesis, coarse alignment via point cloud fitting, environment setup, layout optimization, and ergonomic adjustment using relation-specific differentiable losses.  A closed-loop auto-verification mechanism further ensures fidelity to both semantic constraints and geometric plausibility by continuously monitoring the evolving scene and triggering corrections when discrepancies are detected.

Key contributions include:
- The formulation of spatially contextualized VLMs that act as agents for structured 3D scene generation through a continually maintained and updated spatial memory.
- The design of a dual-component spatial context (scene portrait + scene hypergraph) that supports multimodal integration and relational reasoning for layout planning and ergonomic refinement.
- An agentic generation workflow combining multiple stages—from asset creation to layout optimization—with built-in feedback via auto-verification.
- Demonstrations of strong generalization across input modalities, producing semantically aligned, editable, and physically plausible 3D scenes, while enabling downstream tasks such as interactive editing and path planning.

Experiments show favorable performance compared to state-of-the-art methods in terms of consistency, aesthetic quality, and geometric plausibility, particularly excelling in preserving spatial structure and style from reference images.

**Strengths:**

The core idea of augmenting VLMs with a structured and persistent spatial context is conceptually compelling. This context comprises two key component: the scene portrait and the scene hypergrap, which jointly capture both the semantic content and spatial relationships within a scene in a coherent and interpretable manner. The integration of partial point clouds as geometric priors for individual assets provides a principled mechanism for grounding abstract multimodal inputs into 3D space, effectively bridging the gap between perceptual understanding and generative modeling. Furthermore, the agent-based design, featuring iterative layout optimization and an auto-verification mechanism, ensures that the generated scenes evolve in a semantically consistent and geometrically plausible manner, thereby mitigating severe spatial or functional irrationalities.

**Weaknesses:**

## Problems on Writing:
1. The authors state in Section 4.2: *Due to space limitations, we provide only the ablation study on environment setup in the main paper;additional ablations are included in the supplementary material.* However, the follow-up content includes the ablation study on environment setup and layout planning.
2. Table 1 reports AQ (aesthetic quality) and FP (functional plausibility) under columns labeled “(4o/User)(↑)”, suggesting scores from both GPT-4o and human users. However, the paper does not specify whether these values represent ordinal rankings or normalized scores. Given that lower numerical values indicate better performance (e.g., Ours: 1.00), they appear to be relative ranks—but this must be explicitly stated. Without clarification, quantitative comparisons are ambiguous and potentially misleading.

## Over-Reliance on External Generators Introduces Confounding Factors
The framework heavily depends on external APIs (e.g., Meshy.ai) for 3D asset generation. While practical, this reliance means that aspects of appearance fidelity, texture quality, and stylistic consistency are largely determined by third-party models rather than the proposed method itself. As a result, it becomes difficult to disentangle the contribution of the spatial context mechanism from the inherent capabilities of the black-box generator. For instance, if the final scene lacks style consistency with a reference image, is this due to failure in context modeling or limitations of the mesh generator?

## Insufficient Ablation Study
The paper lacks ablation studies directly evaluating the proposed spatial context. In the ablation study, the effect of the overall replacement with other methods was compared only by graphical comparison. It is unclear how essential each component is to the final performance. This weakens the argument for the necessity of the full spatial context design.

## Limited Discussion of Scalability and Failure Cases
The method assumes accurate Initial point cloud reconstruction and segmentation via existing methods. However, performance likely degrades in cluttered scenes or with small/occluded objects, as briefly acknowledged in the appendix (Limitation A). Yet, there is no empirical analysis of how robust the pipeline is under such conditions. Including failure case such as misaligned furniture due to noisy depth estimation or incorrect hyperedges would strengthen the paper’s credibility and guide future improvements.

**Questions:**

**1. Ablation Study Discrepancy:**
The paper claims only the environment setup ablation is included due to space limits, yet Figure 9 presents a full layout planning ablation. Please clarify this inconsistency and confirm whether additional ablations should exist in the supplement.

**2. Metric Definition:**
Table 1 reports AQ and FP scores (e.g., 1.00 for Ours). Please explicitly define the scoring protocol.

**3. Component Contribution:**
Can you provide evidence isolating the impact of the spatial context (e.g., w/o hypergraph or portrait)?

**4. Failure Cases:**
Could you share one or two representative failure cases to better illustrate current limitations?

---

> ### Author Response · Authors · 2025-12-02
> **Response to Reviewer fJvS**
>
> We thank the reviewer for the constructive and helpful feedback. We have incorporated your suggestions into the revised manuscript, and we address each point below.
>
> **Writing clarity (W1, Q1, Q2).** Section 4.2 has been revised to correct the wording around the ablation discussion and to explicitly define the AQ/FP ordinal ranking protocol. We have also added Table 2, which provides a concise component-level ablation addressing the previously missing details.
>
> **External generators (W2).** The asset generator is a fully modular component and can be replaced without affecting the spatial reasoning mechanism. Section 3 now clarifies this separation: appearance quality (style, texture) is determined by the external generator, while object semantics, placement, and relational consistency are governed by our spatial context.
>
> **Additional ablation (W3, Q3).** To better isolate the contributions of each part of the spatial context, we now include a compact component-level ablation (Table 2) analyzing the roles of the portrait, hypergraph, and auto-verification.
>
> **Scalability and failure cases (W4, Q4).** The revised version includes several representative failure cases (e.g., scenes with noisy depth, small or heavily occluded objects) to provide a more transparent view of the current limitations and to contextualize scalability concerns.

---

### Official Review · Reviewer_uRi3 · 2025-11-01

**Soundness:** 2
**Presentation:** 3
**Contribution:** 3
**Rating:** 4
**Confidence:** 4

**Summary:**

This paper introduces a pipeline that integrates Vision-Language Models (VLMs) with 3D scene generation technology. By leveraging multimodal inputs, such as images, text, and unstructured image collections, it enables the creation of 3D scenes that align with textual semantics or image content. Additionally, the paper presents an automatic verification mechanism that utilizes VLMs for point cloud restoration throughout the scene generation process. The paper is well-motivated, logically organized, and effectively demonstrates 3D reconstruction results.

**Strengths:**

1. The paper proposes using a structured Spatial Context to provide detailed descriptions of the environment to be generated. This structured data serves as input to the VLM, guiding the 3D scene generation process.
2. During the 3D scene generation, the VLM continuously reads from the Spatial Context, ensuring that the scene remains semantically coherent and geometrically accurate.

**Weaknesses:**

1. In the initialization stage, the pipeline employs Fast3R to construct an initial 3D scene from an image. However, the generated scene contains occlusion relationships between instance assets and the background. Although the paper mentions repairing and generating the invisible regions of the assets, no corresponding repair is conducted for the background. How is the reconstruction completeness of the occluded background areas ensured? According to the supplementary videos, this method can produce a complete scene, especially in cases with only text input. Nevertheless, it remains unclear how the scenes and assets observed from novel viewpoints are generated and aligned with the initialized scene.
2. During instance asset generation, the pipeline first evaluates the completeness of the initialized asset point clouds, but does not present the specific evaluation algorithm or its implementation details. After repairing the assets into complete point clouds, they render front-view images of the assets and combine them with textual descriptions to jointly guide 3D asset generation. In this multimodal guidance process, is priority given to image consistency or semantic consistency? Furthermore, does the incorporation of textual descriptions introduce inconsistencies between the generated 3D assets and the rendered views?
3. In the quantitative comparison experiments, the paper does not disclose the specific settings of the experimental data. For instance, how many scenes were generated in the experiments? How were these scenes rendered, and how many images were rendered to compute metrics such as CLIP?
4. The authors also perform ablation studies on the environment setup, but the paper lacks explanations regarding the specific environmental configuration methods or experimental details. In the supplementary video, the water flow in the environment is dynamic; however, according to the paper, the proposed method does not support the generation of dynamic point cloud data. How is this dynamic scene achieved?
5. In the qualitative experiments and supplementary materials, the authors present only a limited number of results. It would be desirable to include more qualitative examples and report what is the success rate of the proposed method.
6. In Section 3.2, the authors claim that “our framework performs agentic scene generation, where the VLM synthesizes 3D assets.” Given that the output of a VLM is typically textual, the authors should clearly explain the process of asset synthesis.

**Questions:**

The pipeline utilizes a Vision-Language Model (VLM) agent to facilitate 3D scene generation and supports multiple input modalities (text, single image, and image group). However, it does not specify the experimental protocol adopted when using an image set as input. Based on the qualitative examples in Figure 8, this image set appears to include both asset and scene images. If multi-view scene images are used instead, would the experimental protocol differ?

---

> ### Author Response · Authors · 2025-12-02
> **Response to Reviewer uRi3**
>
> We sincerely thank the reviewer for the constructive feedback. Your suggestions have strengthened the revision, and we address all specific points in the following responses.
>
> **Background completeness (W1).** The system does not reconstruct the background from a single input view, as large portions are fundamentally unobservable. Missing regions are handled in the environment setup stage, where ceiling, wall surfaces, terrain, and water surfaces are generated procedurally from the portrait.
>
> **Completeness evaluation and multimodal guidance (W2).** There is no separate completeness-evaluation algorithm; all assets are passed through the restoration module. After restoration, the canonical front view provides geometric cues, while the textual sub-description provides high-level semantics for asset generation.
>
> **Evaluation protocol clarity (W3).** Section 4.2 has been revised to explicitly specify the prompt set, rendering protocol, number of images per scene, and the exact computation procedures for CLIP/BLIP/LPIPS. The ranking protocol used for human and GPT-4o evaluations is now clearly stated.
>
> **Environment setup (W4).** Environment elements—walls, ceiling, terrain, water body—are generated procedurally based on the portrait. Dynamic water in the video comes from Blender’s built-in procedural animation applied after geometry generation.
>
> **Qualitative results and success rate (W5).** The supplementary material includes representative cases across modalities. Because empirical “success rate” is subjective, the revised version will instead provide several typical failure cases to illustrate limitations.
>
> **Asset synthesis process (W6).** The VLM outputs refined textual descriptions, not geometry. Each description is paired with the canonical front view of the restored point cloud and passed to the external asset generator. Section 3.2 has been updated for clarity.
>
> **Image-set protocol (Q1).** The protocol is unchanged for image sets. Object-centered, scene-level, or multi-view images are merged into the portrait and jointly interpreted by the VLM. The agentic loop operates on the unified representation without changing later steps.

---

### Official Review · Reviewer_hRfo · 2025-11-02

**Soundness:** 3
**Presentation:** 2
**Contribution:** 2
**Rating:** 4
**Confidence:** 3

**Summary:**

This paper proposes a framework for agentic 3D scene generation by augmenting vision-language models (VLMs) with a spatial context composed of two key components: a scene portrait (semantic layout blueprint) and a scene hypergraph (object–object and object–environment relations). The method enables a VLM (GPT-4o) to act as an autonomous agent that iteratively constructs, verifies, and refines 3D environments from text, images, or unstructured image sets. The system performs asset synthesis, layout planning, ergonomic adjustment, and environment setup, while an auto-verification module ensures geometric and semantic consistency. Experiments demonstrate strong performance compared to Holodeck, DreamScene, and ACDC, achieving better alignment and plausibility across text-, image-, and multi-view inputs.

**Strengths:**

1. the paper’s notion of “spatially contextualized VLMs” is original and well-motivated. It reinterprets VLMs as reasoning agents operating over structured 3D contexts.
2. The ability to handle both single images and unstructured image collections is impressive and practically relevant.

**Weaknesses:**

1. reported metrics (CLIP, BLIP, LPIPS) evaluate rendered 2D projections. There is no quantitative evaluation of 3D accuracy, geometry consistency, or spatial relation correctness—critical aspects for a 3D generation paper.
2. while conceptually coherent, the pipeline involves multiple submodules (Fast3R, Point-M2AE, Meshy, Blender), which could make it hard to scale or analyze systematically.
3. while the conceptual framing of “spatially contextualized VLMs” is interesting, many subcomponents: scene graphs, point-cloud restoration, and ergonomic layout optimization, build directly on existing methods (e.g., ATISS, Point-M2AE, Fast3R). The integration is thoughtful but primarily a system-level composition rather than a new algorithmic breakthrough.
4. the paper does not clearly define what datasets or prompts were used for quantitative evaluation or how they were standardized across methods. Without standardized benchmarks, numerical comparisons (especially involving user studies) may be subjective.

**Questions:**

1. do you provide an object level promp (an electric guitar), or you extract it from the main promps? can you do better when given more detailed object level prompt?
2. How does StructMap handle self-occlusion when rendered to 2D?
3 the first image can be generated using the prompt? if there are any missing object between the prompt and the generated image, will your approach work?

---

> ### Author Response · Authors · 2025-12-02
> **Response to Reviewer hRfo**
>
> We thank the reviewer for the constructive feedback. We appreciate your suggestions and have updated the manuscript accordingly.
>
> **Metrics (W1).** The field currently has no agreed or standardized protocol for measuring free-form 3D scene generation, especially in open-vocabulary, multi-object settings without ground-truth geometry. For comparability, existing work also relies on 2D-rendered metrics (CLIP, BLIP, LPIPS). The revised Section 4.2 introduces simple 3D validity measures, collision rate and support-violation rate, that evaluate spatial plausibility and directly reflect the contribution of our spatial context.
>
> **Pipeline complexity and scalability (W2).** All submodules such as depth estimation, point-cloud restoration, asset generation, and rendering can be replaced without affecting the spatial-context reasoning. The core mechanism remains the agentic loop operating over the scene portrait and hypergraph, keeping the system analyzable and scalable regardless of specific tool choices.
>
> **Algorithmic novelty (W3).** Spatially contextualized VLMs are defined by the unified spatial context and the iterative agentic loop that continuously reads, updates, and verifies the scene. The portrait–hypergraph representation functions as a semantic–geometric working memory not present in prior work. Together with the closed-loop auto-verification, the novelty lies in the representation and control paradigm rather than re-implementing existing modules.
>
> **Evaluation protocol clarity (W4).** The revised version now provides a complete evaluation protocol. We explicitly report the prompt set, rendering configuration, and metric computation for CLIP/BLIP/LPIPS and the human/GPT-4o ranking procedure.
>
> **Object-level prompts (Q1).** The system does not depend on user-provided object-level prompts. The VLM derives object descriptions from the main prompt, from semantic cues in the portrait, and from the canonical front-view projection of the restored point cloud. Section 3.2 now clarifies this. Extra detailed descriptions, when provided, are naturally incorporated and often improve fidelity.
>
> **Self-occlusion handling (Q2).** StructMap operates on the restored point cloud, not raw depth. Occluded or missing regions are completed using our Point-M2AE module, and the completed geometry is used for all 2D projections in verification, avoiding self-occlusion artifacts.
>
> **Missing objects between prompt and image (Q3).** If an object is mentioned in the prompt but not present in the image, the textual description in the portrait is used to generate and place it. Hypergraph relations determine its placement, ensuring prompt consistency even with incomplete observations.

---

### Meta-Review · Area_Chair_MyNS · 2026-01-02

**Summary:**

This paper proposes an agentic framework for 3D scene generation that augments vision-language models with a structured spatial context, including a scene portrait and a scene hypergraph. Reviewers generally find the high-level idea interesting and the system well-motivated, but raise consistent concerns about limited algorithmic novelty, heavy reliance on existing components and external generators, and insufficiently grounded evaluation for a 3D scene generation paper. While the rebuttal provides clarifications and additional analyses, these responses do not fully address the core concerns regarding contribution depth and empirical rigor. The AC recommends rejection.

**Reviewer Concerns:**

Reviewers raise concerns primarily around the nature of the contribution and the strength of the evaluation. While the framework integrates multiple components into a coherent pipeline, much of the methodology builds on existing representations, reconstruction tools, and external asset generators, making the contribution largely system-level rather than algorithmic. Evaluation relies heavily on 2D rendered metrics and subjective rankings, with limited quantitative validation of 3D geometry, spatial relations, or component-level necessity. Additional issues include insufficient ablations to isolate the impact of the proposed spatial context, unclear scalability and robustness in complex scenes, and ambiguities in experimental protocols. Although the rebuttal addresses several presentation and clarity issues, the main concerns about novelty and evaluation remain outstanding.

**Reviewer Scores:**

- Reviewer hRfo: Initially assigns a score of 4, citing interesting conceptual framing but weak 3D evaluation and limited algorithmic contribution. After the rebuttal, the core concerns remain, and the score would likely remain at a similar below-threshold level (4).
- Reviewer uRi3: Initially assigns a score of 4, noting unclear evaluation protocols, limited ablations, and questions about the asset synthesis process. Despite clarifications in the rebuttal, the assessment would likely remain unchanged (4).
- Reviewer fJvS: Initially assigns a score of 6, acknowledging conceptual appeal but expressing concerns about reliance on external generators, insufficient ablations, and unclear metric definitions. After the rebuttal, the score would likely decrease slightly or remain borderline (around 6), but not shift to clear acceptance.
- Reviewer wXxD: Initially assigns a score of 4, raising concerns about the fairness of comparisons, dependence on upstream models, and limited qualitative evidence. These concerns are partially clarified but not fully resolved, and the score would likely remain at a similar level (4).
- Reviewer mwhr: Initially assigns a score of 4, highlighting unclear novelty relative to prior scene-graph-based methods and limited ablation support. After the rebuttal, the score would likely remain unchanged (4).

---

### Decision · Program_Chairs · 2026-01-26

Reject